# Identification of Huge Phages from Wastewater Metagenomes

**DOI:** 10.3390/v15122330

**Published:** 2023-11-28

**Authors:** René Kallies, Die Hu, Nafi’u Abdulkadir, Michael Schloter, Ulisses Rocha

**Affiliations:** 1Department for Environmental Microbiology, Helmholtz Centre for Environmental Research, Permoserstr. 15, D-04318 Leipzig, Germany; d.hu@ufz.de (D.H.); nafiu.abdulkadir@ufz.de (N.A.); 2Department of Environmental Health, Helmholtz Munich, Ingolstaedter Landstr. 1, D-85758 Neuherberg, Germany; schloter@helmholtz-muenchen.de

**Keywords:** huge phage, wastewater, metagenome, viral metagenomics, virus genome annotation, virus phylogeny

## Abstract

Huge phages have genomes larger than 200 kilobases, which are particularly interesting for their genetic inventory and evolution. We screened 165 wastewater metagenomes for the presence of viral sequences. After identifying over 600 potential huge phage genomes, we reduced the dataset using manual curation by excluding viral contigs that did not contain viral protein-coding genes or consisted of concatemers of several small phage genomes. This dataset showed seven fully annotated huge phage genomes. The phages grouped into distinct phylogenetic clades, likely forming new genera and families. A phylogenomic analysis between our huge phages and phages with smaller genomes, i.e., less than 200 kb, supported the hypothesis that huge phages have undergone convergent evolution. The genomes contained typical phage protein-coding genes, sequential gene cassettes for metabolic pathways, and complete inventories of tRNA genes covering all standard and rare amino acids. Our study showed a pipeline for huge phage analyses that may lead to new enzymes for therapeutic or biotechnological applications.

## 1. Introduction

Bacteria-infecting viruses, or phages, are extremely diverse and present in all ecosystems studied to date. Furthermore, they hold significant ecological importance, as they can lyse their hosts, facilitate horizontal gene transfer, and modify host metabolism, thereby exerting a pivotal role in shaping microbial community structures. [1,2,3,4]. Metagenomics studies the genetic reservoir in diverse environmental samples [5,6,7,8] and helps to identify the (microbial) genomes in such samples [9]. In addition to small phages, phages with DNA genomes larger than 200 kb in genome size have recently received more attention [10] since *Bacillus megatherium* phage G was described in 1973 as the first large prokaryotic virus with a head-to-tail length of 600 nm and a diameter of about 200 nanometres [11]. This ‘prototype’ huge phage has a genome size of just under 500 kilobases [12]. Such huge phages are known by different names, such as ‘jumbophages’ [13] or also ‘megaphages’ for phages with a genome size of more than 500 kb [14]. Recently, however, it has been proposed to refer to such phages simply as ‘huge phages’ [10]. Huge phages show high diversity, infect hosts of different bacterial phyla, and differ from smaller phages in genome organisation and gene expression patterns [13]. Huge phages have been isolated or identified from various environmental systems, but most of them have been identified from an aquatic environment, presumably because they can infect their hosts more effectively there as they can diffuse more easily [13,15]. Other environments for large phages include soil, sediments, plants, and animal guts [10,13]. Recently, Prevotella-infecting phages have been identified in the human gut [14], suggesting that these phages are widespread. Large phages have interesting morphological and genomic features. They have both contractile and non-contractile tails, suggesting different evolution in different phage groups [16]. Their heads and tails include structural variations such as fibres attached to the heads, as shown for the *Tenacibaculum maritimum* phages PTm1 and PTm5 [17], and variations in the tail fibre morphology [18,19,20]. Huge phages have some unique genomic and biological properties. For example, all huge phages have DNA polymerases belonging to different DNA polymerase types, indicating that their replication is independent of the host [16]. Other features include the presence of genes for tRNA modification, genes for proteins that influence or take over host translation, such as initiation factors, or unique CRISPR systems [10,21]. Furthermore, huge phages can synthesise their own NAD+, which is required as a source for DNA replication and the regulatory enzymes of the phages [22]. Huge phages are also known to have several tRNA genes, allowing them to evade host defence mechanisms [10,13]. They are also interesting from evolutionary and ecological perspectives. For example, it has been proposed that these phages have evolved from smaller phages and have developed a k-strategy rather than an r-strategy for reproduction [23]. In addition, it has been suggested that these huge phages are ancient, having evolved together with free-living cells and their symbionts from a common primordial ancestor and having developed their replication strategies [10]. It should be noted that huge phages must have more genetic information than their smaller counterparts simply by virtue of their larger genomes [13]. For example, phage enzymes have become the targets of therapeutic or biotechnological applications [24,25,26], so it seems worthwhile to study the genetic and, thus, functional potential of these huge phages.

However, the analysis of huge phages was limited for a long time due to missing analytical tools. For example, the isolation of these phages is limited because they may have difficulty diffusing in the medium and, therefore, do not form plaques [27] or are removed via filtration during the methodological process [13]. The analysis has recently advanced due to the possibility of using large metagenomes containing significant (unknown) genetic information and offering the unique potential to identify novel viruses, including huge phages. However, their assembly from raw sequencing reads and the analysis can be challenging [10,28], requiring improved bioinformatics pipelines. Based on advancements in this field, huge phages have been identified from large metagenomes and isolated from different environments [10,14,29,30,31] in the past few years.

This work focused on identifying huge phages from 165 wastewater metagenomes available in public databases [5]. Wastewater represents a fingerprint of human and environmental microbiota and thus may contain a significant number of highly diverse huge phages [32]. Here, we describe in detail seven huge phages filtered from an initial dataset of more than 1.5 million putative viral sequences, demonstrating that efforts in improving bioinformatical pipelines are still required to identify phage genomes of interest from larger metagenomic datasets.

## 2. Materials and Methods

### 2.1. Metagenome Dataset

A total of 6000 curated metagenomes were collected from the TerrestrialMetagenomeDB [5] within the Collaborative Multi-domain Exploration of Terrestrial metagenomes (CLUE-TERRA) consortium (https://www.ufz.de/index.php?en=47300, accessed on 23 May 2023) as described elsewhere [33]. From these, metagenomes with the keywords ‘activated sludge’ and ‘wastewater’ were selected, resulting in 165 metagenomes that were further analysed in this study. Of these metagenomes, 66 were from Asia, 4 were from Asia, 47 were from Europe, 34 were from North America, and 14 were from South America. An overview of the metagenome libraries is provided in Appendix A. The metagenome libraries consisted of short read sequences with an average library fragment length of 150 to 602 bases (median 302 bases) (Appendix A).

### 2.2. Virus Sequence Identification

We used the Multi-Domain Genome Recovery v1.0.1 pipeline to identify viral contigs from the 165 metagenome libraries [34]. Briefly, VirSorter 2 v2.2.4 [35], VirFinder v1.1 [36], and VIBRANT v1.2.1 [37] were used with default settings to identify viral contigs from assemblies that were generated with Spades 3.15.2 [38]. Repeated sequences (from contigs identified by two or three tools) were removed, and putative viral contigs were then dereplicated to a 95% average nucleotide identity over at least 70% of the shortest sequence. The completeness and quality of the contigs were then checked using CheckV 1.0.1 [39].

As this study aimed to identify huge phage genomes, we continued our analyses with complete and high-quality viral genomes of more than 200 kb in length.

### 2.3. Genome Manual Curation

Contigs were first checked for circularisation. This was carried out by reference mapping sequencing reads from the respective library to the contig of interest using the Geneious reference mapping tool. The mapped reads were then *de novo* assembled using the ‘circularise’ option implemented in the Geneious Prime^®^ 2023.0.4 *de novo* assembler (https://www.geneious.com, accessed on 15 June 2023). All circularised contigs were considered complete. All contigs of interest were checked for erroneous concatenation. These contigs consist of at least two sequences from different viruses or the same virus due to, e.g., assembled terminal repeat regions. VIBRANT’s machine learning-based neural network helped us to identify such contigs initially. In addition, repeat regions >3 kb were identified using Vmatch (http://www.vmatch.de/, accessed on 17 June 2023), Geneious Repeat Finder, and dot blot analysis. Reference mappings were used to fill gaps and extend ends where necessary. Single and small stretches of Ns and ambiguities were identified via contig self-alignment and manually curated using reference mappings.

### 2.4. Annotation of Structure, Functional Potential, and Lifestyle

Genes and coding sequences (CDS) in the curated phage genomes were predicted using a combination of Prodigal [40] and PHANOTATE [41] implemented in the VIBRANT and Pharokka [42] pipelines. Functional annotation of translated CDS was performed through the search against the PHROG [43], CARD [44], and VFDB [45] databases using MMseqs2 [46] and against the KEGG (release 105.0) [47], PFam (v32) [48], and VOGs (release 94) [49] databases. The annotation was improved by BLASTp [50,51] alignments against the NCBI non-redundant database [52] and HHPred searches [53] against the Conserved Domain Database [54], COG database [55], and UniProt-swiss-viral [56] database. Genomes were scanned for tRNAs and tmRNAs using tRNA-scan SE2 [57,58] and Aragorn [59]. CRISPR loci were identified with CRISPRCasFinder [60,61]. Genome orientation was checked via the orientation (positive or negative strand) of the terminase large subunit. If necessary, genomes were reoriented to begin with the large terminase subunit gene in positive orientation using the Pharokka re-orientation mode. The lifestyles of identified phages were predicted using PhaTYP [62].

### 2.5. Genome Comparison and Phylogenetic Analysis

The average nucleotide identity (ANI) of the seven phage genomes to the known phage genomes was determined using the OrthoANIu tool [63]. For this purpose, a total of 21,217 complete bacterial virus genomes available from Genbank on 18 July 2023 and from a recent study [10] were used.

The major capsid protein (MCP) is one of the most conserved proteins encoded by bacteriophages and has been widely used for phylogenetic analyses [64]. The MCP gene nucleotide sequences were translated into amino acid (aa) sequences, and related sequences available in GenBank were identified using PSI-BLAST [51] against both the viral non-redundant and the viral RefSeq databases [52]. A maximum of 10 hits were allowed, and the corresponding genome sequences were downloaded. Several huge phage genomes have recently been described [10]. The MCP genes of these genomes were validated and corrected via a comparison with known homologous genes and included in the analysis. Amino acid alignments were performed using MAFFT v7.490 (scoring matrix: BLOSUM62, gap open penalty: 1.53, offset value: 0.123) [65]. The alignment was trimmed using trimAL [66] with gappyout settings. The maximum likelihood phylogenetic tree was constructed with IQ-TREE 1.6.12 [67] using automatic model selection [68] and ultrafast bootstrap [69] option with 1000 bootstrap replicates. The resulting consensus tree was rooted in Herpes simplex virus 1 MCP that was used as an outgroup and visualised in iTOL [70].

A set of phage genomes was used for genome-based phylogeny using VICTOR (https://victor.dsmz.de, accessed on 28 June 2023) [71]. The set included genomes related to our genomes based on the ANI and the MCP phylogeny and one representative genome from each of the 20 proposed clades from a recent study [10]. All pairwise comparisons of nucleotide sequences were performed using the Genome-BLAST Distance Phylogeny (GBDP) method [72] with settings recommended for prokaryotic viruses [71]. The resulting intergenomic distances were used to infer a balanced minimum evolution tree with branch support via FASTME, including SPR post-processing [73] for the formula D0. Branch support was inferred from 100 pseudo-bootstrap replicates. Trees were midpoint-rooted [74] and visualised using ggtree [75]. Taxon boundaries at the species, genus, and family levels were estimated using the OPTSIL program [76], the recommended clustering thresholds [71], and an F-value (fraction of links required for cluster fusion) of 0.5 [77].

For confirmation, genome sequences of the same dataset were used to generate a ‘proteome-wide’ tree with the ViPTree server [78]. The dendrogram was based on genome-wide sequence similarities computed using tBLASTx [50].

Default settings were used for all tools.

## 3. Results

### 3.1. Genome Identification and Major Features

We identified a total of 2,578,604 (per library, min: 141; max: 52,046; median: 10,828) dereplicated putative viral contigs from the 165 wastewater metagenomic libraries, of which 12,337 contigs were predicted to be provirus sequences. We applied CheckV to estimate genome completeness and quality and identified 684 complete and 1099 > 90% complete (i.e., CheckV high-quality category) contigs, while the remaining contigs had less than 90% or unknown completeness (Appendix A). We then filtered the contigs by size, selecting only those larger than 200 kb and with CheckV qualities of ‘Complete’, ‘High-quality’, and ‘Medium-quality’. This selection reduced the dataset to 7 complete, 154 high-quality, and 36 medium-quality contigs. The contig sizes ranged from 201 kb to 1684 kb (median: 295 kb) (Appendix A).

The contigs were then screened for viral hallmark genes, and only contigs containing at least terminase or structural protein genes (e.g., capsid, portal, tail, baseplate) were retained for further analysis. The contigs were also screened for ribosomal genes (i.e., coding for 16S and 23S ribosomal subunits), and contigs containing such genes were excluded. These screens were especially true for the very large contigs. Afterwards, the contigs were checked for long repeats > 3 kb to exclude concatenated contigs (either via self-concatenation or the concatenation of several different shorter contigs). The final dataset that was used for further analyses, such as annotation and phylogeny, was thus reduced to seven contigs (three with complete CheckV quality scores, two with high-quality scores, and two with medium-quality scores). These seven contigs were identified from four sequencing libraries: 1-SewaA from an activated sludge wastewater sample from Japan (sample ID: EADRX012718); 2-SewaB and 3-SewaC from activated sludge from a domestic wastewater treatment plant in Singapore (sample ID: EASRX1759564); 4-SewaD, 5-SewaE, and 7-SewaG from a Japanese activated sludge sample of municipal wastewater treatment plant (sample ID: EASRX2157902); and 6-SewaF from another Japanese municipal wastewater treatment plant (sample ID: EASRX2157911) (Appendix A).

An attempt was made to circularise all contigs to check if they were complete, and this was successful for four of the seven contigs. Therefore, whether the other three contigs are complete or have a linear structure with no clear terminal repeats is unclear. The genome sizes ranged from 204,222 bp to 303,942 bp, with 276 to 544 predicted coding sequences. Between 5 and 47 tRNA genes were found in the genomes (Table 1).

### 3.2. Phylogeny and Taxonomy

The seven phage sequences’ average nucleotide identity (ANI) was compared with 21,217 complete bacterial virus genomes available from Genbank on 18 July 2023 and a previous study [10]. The ANI between the phage sequences from this study and the database sequences was generally relatively low. Most sequences had a query coverage of less than 1 kb to already known sequences. Matches to the NCBI sequences were found for only two of the seven genomes. Most hits were against sequences from the previous study about huge phage genomes mentioned above [10]. The ANI of the known sequences ranged from 56.6% with a 0.27% query coverage (5-SewaE) to 65.3% ANI with a 6.8% query coverage (6-SewaF). No ANI to any database sequence was found for 3-SewaC. A summary of the ANI comparisons can be found in Appendix A. These low similarities to already known phage sequences were supported by genome-wide tblastx analyses (Appendix A) [79].

Proteomic (tbastx-based) and genome-wide sequence similarity-based phylogenies confirmed the above results and placed the seven phage genomes in distinct clades with similarities to huge phages identified in the metagenomic studies [80,81]. Only 7-SewaG shared a most recent common ancestor with the already established Prevotella Lak phage clade (Figure 1 and Appendix A). The OPTSIL software [76] implemented in VICTOR determines the boundaries of species, genera, and (sub-)families. Based on these boundaries, 1-SewaA, 4-SewaD, 6-SewaF, and 7-SewaG would be members of the same virus family, with 4-SewaD, 6-SewaF, and 7-SewaG being the only known members of different genera, whereas 1-SewaA would form one genus together with an uncultured *Caudoviricetes* phage (GenBAnk Acc. No. LR797474). To test this hypothesis, we calculated the intergenomic similarity of these two phages using VIRIDIC with a 70% genus threshold [82]. The two genomes were 20% identical, suggesting that the two phages are more distantly related than what was suggested by the VICTOR analyses. As the only known members, the phages 2-SewaB, 3-SewaC, and 5-SewaE would form new families based on the VICTOR calculation (Figure 1).

The major capsid protein (MCP) is a viral protein that is widely used for phylogenetic analyses. We performed a blastp search (including iterative psi blast searches) with the seven MCP amino acid sequences against the NCBI non-redundant and viral RefSeq databases to identify related MCP sequences. We identified 99 related MCP sequences from the NCBI database with an e-value of less than 0.01 and included them in the phylogenetic analysis and available MCP sequences from each of the 13 recently proposed representative huge phage clusters [10]. All of these were from unclassified members of *Caudoviricetes*. The herpes simplex virus 1 MCP served as an outgroup. The MCPs of 1-SewaA and 7-SewaG clustered in a clade with uncultured *Caudoviricetes* phages assembled from freshwater metagenomes [80]. The MCPs of 2-SewaB and 3-SewaC had high degrees of divergence from known phages and shared the most common recent ancestors, with a distantly related phage identified from a human metagenome [81] (2-SewaB) and the giant *Bacillus* phage 0305phi8-36 [83] (3-SewaC). The MCP of 4-SewaD clustered in a distinct clade with an uncultured freshwater metagenome-assembled phage [80], and the MCP of 5-SewaE clustered in a related but distinct clade with a recently described giant phage, which was placed in one of twenty newly established huge phage clades called Biggiphage [10] (Figure 2).

### 3.3. Functional Annotation

Using two different phage annotation tools, we could predict 2861 coding sequences (CDS) for all seven phages (276 to 544), of which 81.37% remained hypothetical. Between 11.4% and 26.1% of the CDS could be assigned a function (Table 1, Appendix A).

Most of the proteins (11.4%) with predicted functions include proteins involved in nucleic acid metabolism, such as replication enzymes (polymerases, primases, ligases, and helicases) and enzymes that may be involved in nucleic acid modification and other biochemical processes that are important for phage replication and their interaction with host cells (nucleases, proteases, reductases, and transferases). Structural proteins include phage-typical proteins such as capsid, portal, tail, and baseplate proteins.

The individual phage genomes encode between 1 and 13 predicted Auxilliary Metabolic Genes (AMGs), which can be involved in 23 different signalling pathways. Genes involved in nicotinate and nicotinamide metabolism were very prominent. For example, 5-SewaE encodes four different genes whose gene products are directly involved in converting nicotinamide to NAD+. Another example of such a gene cassette of successive enzymes is found in 3-SewaC. This phage genome contains genes for four enzymes that catalyse dTDP-L-rhamnose production from D-glucose-1P via three intermediates. These enzymes are involved in synthesising polyketide sugars, which are part of the biosynthetic pathways of vancomycin antibiotics, such as streptamycin, or glycosidase inhibitors, such as acarbose or validamycin. Other AMGs encode for enzymes involved in amino acid and nucleotide biosynthesis, such as folate and sulphur relay systems, as well as sugar and vitamin biosynthesis. Another AMG identified in 1-SewaA is the heptosyltransferase I, which attaches heptose sugar units to the lipid A molecule in lipopolysaccharide biosynthesis. Lipid A is the hydrophobic component of LPS, which is closely associated with the outer membrane of Gram-negative bacteria [84]. The addition of heptose sugar units by heptosyltransferase I is critical for forming the characteristic LPS structure, and it plays a role in immune recognition and the bacterium’s interaction with its environment [85,86]. This interaction is another example of how the phages described here can interfere with different metabolic processes in their hosts. A summary of the identified AMGs is presented in Appendix A.

The seven phage genomes contained between 5 and 47 transfer RNA (tRNA) genes are shown in Table 1.

The putative CRISPR sequences were identified in two phage genomes (two cassettes in 5-SewaE and three cassettes in 6-SewaF). Only one of the CRISPR cassettes in 5-SewaE showed a 93% similarity to the known CAS type I-C system sequences (Appendix A, CRISPR sequences).

**Figure 1 viruses-15-02330-f001:**
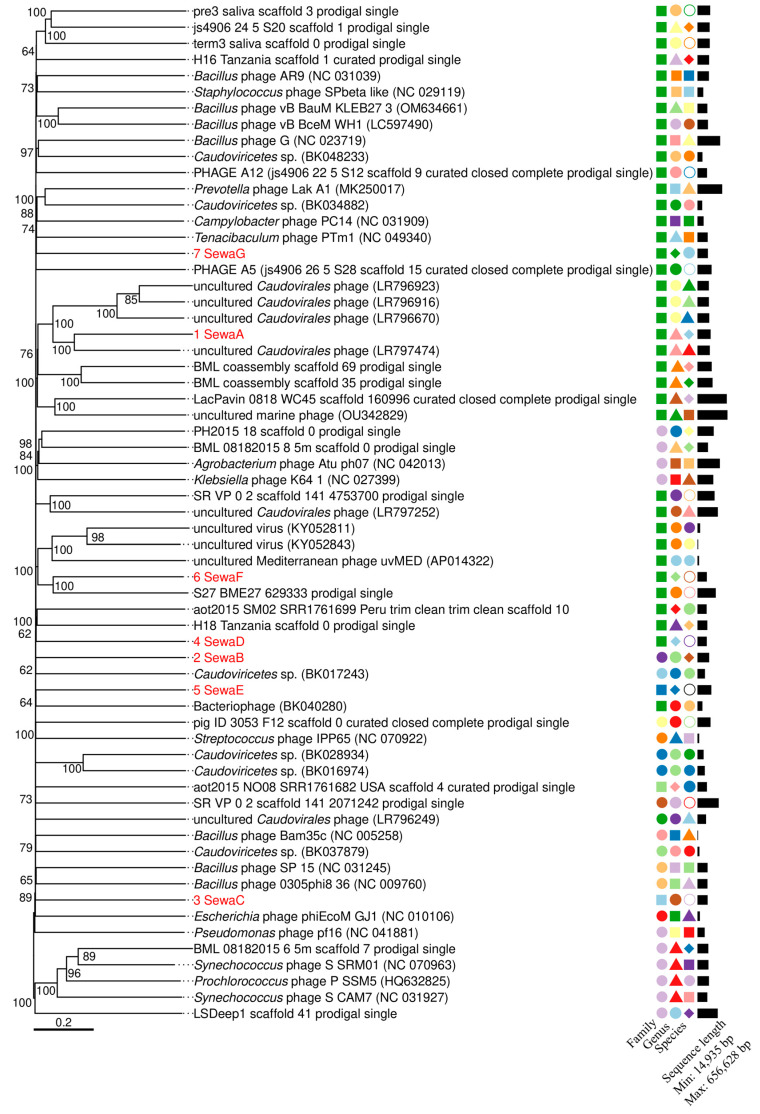
Phylogenomic GBDP tree inferred using the formula D0 (nucleotide) and yielding average support of 64%. The numbers near branches are GBDP pseudo-bootstrap support values from 100 replications. The branch lengths of the resulting VICTOR tree are scaled in terms of the respective distance formula used. The OPTSIL clustering [76] yielded 81 species clusters and 60 genus clusters. The number of clusters determined at the family level was fifteen. The geometrical shapes and colours represent different taxa. Studied phages are shown in red.

**Figure 2 viruses-15-02330-f002:**
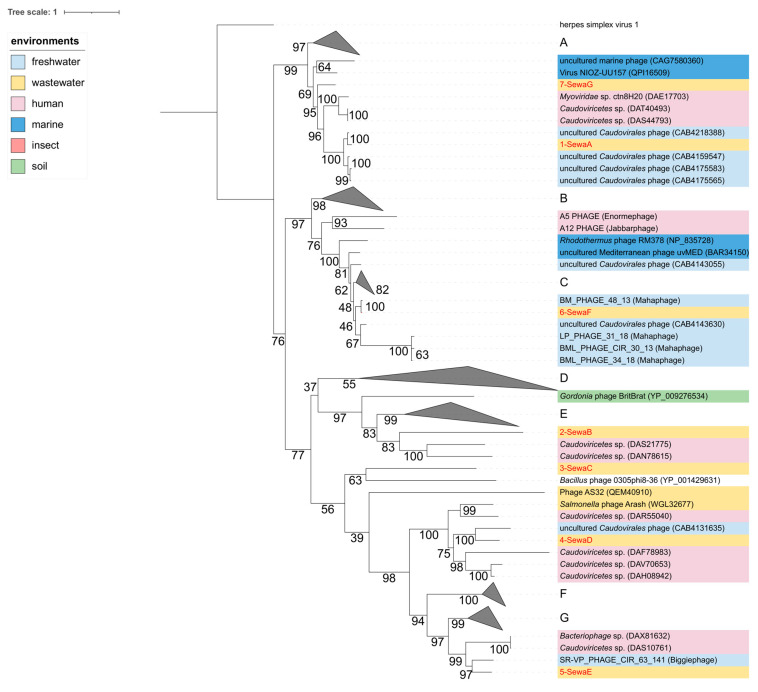
Phylogenetic relationship of identified huge phage major capsid proteins. Maximum likelihood phylogeny based on the amino acid sequences of the major capsid proteins was performed in iqtree with the best model option. Confidence tests were performed with 1000 bootstrap replicates. Virus names are shown, and GenBank accession numbers are given in parentheses. Some clades were collapsed for clarity (**A**–**G**). Coloured backgrounds represent the environment from which the sequences were identified (see legend).

## 4. Discussion

Our work aimed to analyse wastewater metagenomes for the presence of huge phages. We used 165 metagenomes available in public databases. We identified about 2.5 million potential viral sequences using bioinformatic methods and gradually reduced this dataset for a deeper analysis. Since we were interested in phages with a genome size of more than 200 kb, we reduced the dataset to 638 viral contigs of interest. For the quality analysis, we used CheckV, and due to the large number of potentially interesting contigs, we focused on the categories of ‘complete’, ‘high-quality’, and medium-quality, which reduced the dataset to 197 contigs. These contigs were subjected to a more detailed analysis by searching for typical viral protein-coding genes (i.e., hallmark genes). In many of the contigs, none of these genes were identified, so we excluded them from the analysis. We also searched for bacterial genes, particularly genes encoding ribosomal subunits. In this way, we had to exclude other potential viral contigs from further analysis. In particular, these were very large contigs of more than 1 Mb in size. Our study highlights a major problem in identifying and analysing huge phages from metagenomes. This problem represents a challenge due to the large amount of data, selection criteria, quality control, and difficulties in identifying appropriate sequences for which appropriate computational pipelines should be developed.

Another problem was contigs consisting of concatemers of at least two smaller phage genomes. This problem resulted from the incorrect assembly of sequencing reads because terminal repeat regions of phage genomes cannot be distinguished as belonging to individual phage genomes during the assembly process [10]. In this way, we reduced our dataset to seven contigs, which we analysed in detail. This process clearly shows that predicting viruses from metagenomic datasets and calculating the quality of potential viral contigs should be carried out with caution. CheckV is dependent on the available datasets on which the estimates are calculated [39], and like any bioinformatics tool, it has limitations. On the other hand, there is a high probability of more potential contigs of interest (i.e., viral sequences), specifically in the ‘not determined’ category. We deliberately chose not to explore these categories in more detail in this work, as the manual curation and annotation of many large viral genomes were outside the scope of this work. Our data also demonstrate that the existing computational methods cannot analyse large viral datasets with 100% confidence. A manual analysis of hundreds of thousands to millions of potentially viral sequences, such as those typically generated from large metagenomic datasets [81,87,88], is not feasible. Our study encountered challenges in assembling and characterising large phage genomes from metagenomic datasets. It is worth noting that using long-read sequencing technologies could potentially provide more contiguous and accurate genome reconstructions, helping to overcome some of the limitations associated with short-read sequencing approaches.

The phylogenetic analyses grouped the seven phage genomes described here into distinct clades with relatively large distances to their closest known phage relatives, which are often not further described phages found in metagenome datasets. The large genetic distances indicate a great potential for discovery among huge phages. Based on the VICTOR estimates [76], we described phages belonging to previously undescribed genera or families (Figure 1). These analyses show a relationship between some huge phages and a large divergence among different huge phage clades. It is also worth noting that at both the genomic and proteomic levels, some huge phages appear to be related to smaller phages (with genome sizes well below 200 kb). This suggests that huge phage clades have different common ancestors, i.e., they have evolved through convergent evolution. In general, however, these phages are phylogenetically very distant, so this hypothesis will need to be tested in the future by analysing other as yet unidentified genomes. Phylogenetic analyses should therefore also consider smaller phages and not necessarily assume a 200 kb cut-off. A similar approach was proposed in a recent study [16]. In-depth analyses in this direction could help investigate interesting aspects of phage evolution. For example, it is known that bacteriophages can transfer genes between different phages [89]. In addition, huge phages may have enlarged their genomes by adopting genes from smaller phages. One could speculate that this could have happened through lateral gene transfer, where phages take up genes from other phages or bacteria or where two or more phages with different genome sizes are fused to form a phage with a larger genome by infecting a host with several phages simultaneously. These hypotheses are not well supported since huge phages have unique features that are not found in smaller phage genomes [13]. However, thousands of previously unknown phages have also been discovered in various environments, some of which may have large genomes and novel features [90]. Other possible reasons for such relationships would be the introduction of foreign DNA to enlarge the genome [91] or the loss of non-essential or redundant genome sections, reducing the genome size [92]. Therefore, the origin and evolution of huge phages is still an open question that requires more research.

The majority of the coding sequences identified were annotated as hypothetical proteins. The number of hypothetical proteins is generally approximately half to two-thirds of the predicted open reading frames in phage genomes [93]. However, 73.9% and 88.6% are unusually high. Since most hypothetical proteins are expressed during the early stages of phage infection [94,95,96], these results suggest that huge phages encode a large genetic reservoir to take over the host metabolism and express proteins that are mainly involved in replication, transcription, and translation. About half of the identified coding sequences with predicted functions were involved in nucleic acid metabolism, including enzymes that modify molecules (e.g., amidase, dioxygenase, and transferase), DNA-related enzymes (e.g., polymerases, ligases, and helicases), or enzymes that regulate metabolism (e.g., lipase, hydrolysis, and peptidase), supporting this hypothesis. It is conceivable, for example, that huge phages can act very efficiently to use host resources, energy processes for their own reproduction or overcome host defence mechanisms. Three helicase genes have been identified in one of the phage genomes (4-SewaD), and four other genomes have at least two helicase genes. These helicases are good examples of how huge phages may have adapted to various changing environments and hosts through different replication strategies and enzyme diversification.

Auxiliary metabolic genes (AMGs) have been frequently described in phage genomes and are particularly interesting because they provide a toolkit for influencing host metabolism [97,98,99]. Given the genome size of the phages studied, it would not be surprising to identify at least some AMGs in bigger genomes. Our analyses identified between 1 and 13 different AMGs per genome (Appendix A). Different habitats and hosts may explain these differences in numbers [100]. Some of the AMGs that were identified are particularly interesting because they have several sequential genes for one signalling pathway. Phage 3-SewaC, for example, has four genes that convert D-glucose-1P to dTDP-L-rhamnose, which is part of a polyketide sugar pathway that can be used for antibiotic production and that has already been identified in a similar form in the genome of another huge phage [10], which raises the question of whether some of these signalling pathways are conserved in huge phage genomes. Another phage (5-SewaE) encodes four enzymes that are capable of converting nicotinamide to NAD+. This functional potential suggests that this phage could efficiently influence the host’s energy production or redox regulation to provide resources for its replication or to modify the cell environment accordingly [101]. This finding is even more interesting because such signalling pathways are often absent in gut microbes [102]. Human gut phages (like phages found in wastewater) could complement signalling pathways and contribute to the gut ecosystem service. Such enzymes are also of interest for understanding phage–host interactions and biotechnological processes by identifying enzymes that can be used more efficiently for specific applications.

The presence of tRNA genes in phage genomes can compensate for codon usage bias, i.e., the use of abundant codons in phage genomes compensates for compositional differences between the phage and host genomes, and that tRNA gene abundance is linked to phage virulence [103]. Another interesting explanation for the presence of tRNA genes in phage genomes is that phage tRNAs can evade host defence mechanisms that aim for tRNA-depleting strategies [104]. We identified 5 to 47 tRNA genes in the seven phage genomes (Appendix A). The most interesting is the 1-SewaA phage with its 47 tRNA genes, covering all 20 standard amino acids in varying frequencies. In addition, its genome contains tRNA genes for selenocysteine and formylmethionine (fMet). In addition, two tRNA genes are coding for formylmethionine, which may indicate that this phage can better ensure the flexibility, regulation, and evolutionary adaptation of protein biosynthesis and can use different translation strategies in different environments or under different conditions. Such phages are particularly interesting for phage therapy because the presence of all of the necessary tRNA genes not only ensures more efficient replication and host adaptation, but also, thanks to its efficient translation system, can successfully develop immune escape strategies and thus be used effectively to treat infections. For example, phages carrying tRNA genes that specifically recognise and require fMet for translation could be more effective at invading bacterial cells and taking over their protein production, as fMet plays a particularly important role in bacterial protein synthesis. This knowledge could eventually be used to design customised phages. Furthermore, three of the phage genomes contain tRNA genes for selenocysteine, suggesting that these phages can encode proteins containing selenocysteine and have specific translation systems to regulate selenocysteine incorporation.

Although host analyses were not performed in this study, it is important to consider the potential ecological significance of the seven huge phages identified. These phages may interact with bacterial hosts in WWTPs or originate from human-associated sources. Understanding their potential hosts and roles in these environments could shed light on their contribution to ecosystem resilience and function. Future research in this direction could explore these phages’ ecological niches and interactions and reveal their broader ecological implications.

## 5. Conclusions

We identified more than 600 putative huge phage genome sequences from wastewater metagenomes and described the potential pitfalls of genome analysis. We recommend that the results of computational methods for predicting viral sequences from metagenomes be critically questioned, as many of these sequences are not clearly viral, and the analysis pipelines cannot unambiguously separate concatemers. One proposed solution is to use this knowledge to develop (semi-)automated tools to accurately and rapidly identify actual viral genomes. We described seven huge phage genomes belonging to previously undescribed viral genera and families. These phage genomes contained a repertoire of genes, including cassettes of metabolic genes and complete sets of tRNA genes. This genetic diversity could open up new avenues for biotechnological and medical research. For example, the metabolic genes could be used in the biotechnological production of valuable compounds or drugs, while the tRNA gene sets could serve as the basis for targeted gene expression and the development of therapies. These findings expand our understanding of virological resources and could have significant applications in various fields. Based on a phylogenomic analysis, we also hypothesised that convergent evolution evolved these huge phages. This hypothesis suggests that although these phages belong to different genera and families, they may have adapted and evolved their large genome sizes as an advantageous strategy in their specific ecological niches. Further research and comparative genomic studies may shed more light on convergent evolution.

## Figures and Tables

**Table 1 viruses-15-02330-t001:** Genome feature of huge phage genomes.

Contig Name	Sample ID	Contig Length	Topology	No. of ORFs	No. of ORFs with Annotation ^1^	No. of tRNAs	GC Density	Lifestyle
1-SewaA	EADRX012718	288,455	circular	414	108	47	41.9%	virulent
2-SewaB	EASRX1759564	256,464	circular	544	74	12	36.4%	virulent
3-SewaC	EASRX1759564	222,908	linear	276	44	5	38.7%	virulent
4-SewaD		204,222	circular	323	69	23	36.3%	virulent
5-SewaE		303,942	circular	533	128	28	41.5%	virulent
6-SewaF		205,999	linear	371	57	38	33.6%	virulent
7-SewaG		228,454	linear	402	46	17	30.9%	prophage

Abbreviations: ORF: open reading frame; tRNA: transfer RNA; GC: guanine–cytosine. ^1^ ORFs with assigned function.

## Data Availability

The annotated genome sequences are available in the Appendix A.

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
