# Peer review of "Identification of Huge Phages from Wastewater Metagenomes"

_viruses, 2023, doi:10.3390/v15122330_

Round 1

Reviewer 1 Report

Comments and Suggestions for Authors

This manuscript describes the discovery of 7 large [larger than 200 kb genomes] phages from 165 wastewater metagenomes in the databases. Using a variety of computer programs, they are convinced that the 7 genome sequences really represent new phages.  The new phages fell into distinct phylogenetic clades, likely forming new genera and families. The work is carefully done and the paper is well written. My only comment is based on lines 313-314 - the authors should state in the Materials and Methods section or the beginning of the Results section that all the sequences are from small reads [at least I think this is the case] and maybe provide an average size of the reads.

Author Response

Reviewer 1:

My only comment is based on lines 313-314 - the authors should state in the Materials and Methods section or the beginning of the Results section that all the sequences are from small reads [at least I think this is the case] and maybe provide an average size of the reads.

Response:

Dear reviewer, thank you for the positive feedback on our manuscript. We have added a description at the end of section 2.1: "The metagenome libraries consisted of short read sequences with an average library fragment length of 150 to 602 bases (median 302 bases) (Table S1)." A detailed overview of each library is provided in Table S1.

Reviewer 2 Report

Comments and Suggestions for Authors

The manuscript by Kallies et al. identifies and characterizes huge phage genomes based on metagenomic mining. Generally, the experimental design is clear. The methods used are reasonable. This manuscript updates the catalog of huge phage genomes in wastewater environments. It is an interesting paper, and their findings were valuable. However, several points must be clarified and addressed as presented below.

The introduction part is too brief. The authors should summarize existing research on huge phages and provide more introduction on the characteristics of huge phages to tell the readers the relevance of this topic.

The discussion section needs to be largely improved. The current version repeats the result section too extensively, just like a summary of the results without (or slightly) discussing the results obtained.

L12. ‘excluding’ instead of ‘predicting’ would be better.

L14. ‘demonstrated that reconstruction and analyses of viruses from metagenomes should be carried out cautiously’. No result in this study can support this claim. And it is not the main point of this study. It would be better to remove it.

L31. ‘phages’ rather than ‘pages’.

L147. Keep the value format consistent.

L149. Should be: ‘12,337 contigs were predicted to BE provirus sequences’ rather than contain provirus sequences.

L219. Citation needed.

L221–L225. Any figures or data showing these results?

L262–L265. Citations.

Author Response

Reviewer 2:

The manuscript by Kallies et al. identifies and characterizes huge phage genomes based on metagenomic mining. Generally, the experimental design is clear. The methods used are reasonable. This manuscript updates the catalog of huge phage genomes in wastewater environments. It is an interesting paper, and their findings were valuable. However, several points must be clarified and addressed as presented below.

Response:

Dear reviewer, thank you for critically reading our manuscript. We are grateful for your comments, which helped us to improve our study. Please find below a point-by-point response.

Comment:

The introduction part is too brief. The authors should summarize existing research on huge phages and provide more introduction on the characteristics of huge phages to tell the readers the relevance of this topic.

Response:

We have expanded the introduction to give the reader more information about these phages' habitats, morphological and biological/genomic properties. As a result, we have expanded the citations. We hope this will meet your expectations.

Comment:

The discussion section needs to be largely improved. The current version repeats the result section too extensively, just like a summary of the results without (or slightly) discussing the results obtained.

Response:

We removed some parts from the Results section and included them in the Discussion to clarify the two sections. We have also adapted some parts of the Discussion to make it more specific. However, the manuscript's focus is not a detailed description of the individual phages but rather to report on the challenges of identifying and analyzing such phages. Please see also reviewer 3's comment.

Comment:

L12. 'excluding' instead of 'predicting' would be better.

Response:

We have changed this in the manuscript accordingly.

Comment:

L14. 'demonstrated that reconstruction and analyses of viruses from metagenomes should be carried out cautiously'. No result in this study can support this claim. And it is not the main point of this study. It would be better to remove it.

Response:

This statement referred more to a general statement about viral contigs identified from metagenomes, e.g. when considering the CheckV categories 'complete' and 'high-quality'. However, it is correct that this study does not represent a dataset that would support this general statement, so we have deleted this half-sentence.

Comment:

L31. 'phages' rather than 'pages'.

Response:

We have changed this in the manuscript accordingly.

Comment:

L147. Keep the value format consistent.

Response:

We changed 'median: 10828' to 'median: 10,828 '

Comment:

L149. Should be: '12,337 contigs were predicted to BE provirus sequences' rather than contain provirus sequences.

Response:

We have changed this in the manuscript accordingly.

Comment:

L219. Citation needed.

Response:

We added the corresponding citation.

Comment:

L221–L225. Any figures or data showing these results?

Response:

There is some evidence from phylogenetic analyses when some of the huge phages formed clades together with smaller phages (Figure 1). For phylogeny, we only included phage genomes that showed similarity to our phages after MCP phylogenetic analysis and ANI comparison. We have softened this statement and formulated it as a hypothesis. We further moved that part from the results to the discussion section to also address your concerns about repetitions in the Discussion. Please find the changes highlighted in the manuscript.

Comment:

L262–L265. Citations.

Response:

We have added references.

Reviewer 3 Report

Comments and Suggestions for Authors

The article “Identification of huge phages from wastewater metagenomes” by Rene Kallies and colleagues reports the results of a bioinformatic analysis of wastewater metagenomes for the presence of viral sequences and presents a detailed algorithm for the consistent recovery of genomes of huge phages. The manuscript contains a very informative discussion showing problems and pitfalls of such analysis, and this discussion is indeed very useful for phage researchers. There are only a few minor notes that can be taken into account.

1. Did you inadvertently forget to include Figure S5 in the Supplementary Materials archive?

Line 30 - Should it be “studies” and “helps”?

Lines 126-127 - was it ultrafast bootstrap? Please clarify the IQ-TREE command line settings

Line 147 - Please use commas in numbers.

Lines 159-160 - Did you have contigs containing both MCP (TerL) and rRNA genes?

Lines 201-202 - “whereas 1-SewaA would form one genus together with an uncultured Caudoviricetes phage (GenBAnk Acc. No. LR797474).” - It would be interesting and quick to check this assumption of VICTOR with the help of  VIRIDIC or orthoANI (70% genus threshold)

Line 210 - Caudoviricetes should be written in italics

Line 215 - Bacillus should be written in italics

Line 242 - 81% of the hypothetical proteins are normal for phages, but this is not a small amount. Have you used HHpred (HHblits) for the annotation procedure?

Line 250 - Please explain the AMG acronym here when you use it for the first time.

Author Response

Reviewer 3:

The article "Identification of huge phages from wastewater metagenomes" by Rene Kallies and colleagues reports the results of a bioinformatic analysis of wastewater metagenomes for the presence of viral sequences and presents a detailed algorithm for the consistent recovery of genomes of huge phages. The manuscript contains a very informative discussion showing problems and pitfalls of such analysis, and this discussion is indeed very useful for phage researchers. There are only a few minor notes that can be taken into account.

Response:

Dear reviewer, thank you for your positive feedback on our manuscript. We appreciate your input to improve the manuscript. Please find below a point-by-point response.

Comment:

Did you inadvertently forget to include Figure S5 in the Supplementary Materials archive?

Response:

Figure S5 can be found in the supplementary PDF: 1_Supplementary_Figures_and_Tables on page 6.

Comment:

Line 30 - Should it be "studies" and "helps"?

Response:

We have changed this in the manuscript accordingly.

Comment

Lines 126-127 - was it ultrafast bootstrap? Please clarify the IQ-TREE command line settings

Response:

Yes. We used ModelFinder analysis with ultrafast bootstrap option. We added this information to the text. In addition, we included the corresponding citation for ultrafast bootstrap (Subha Kalyaanamoorthy, Bui Quang Minh, Thomas KF Wong, Arndt von Haeseler, and Lars S Jermiin (2017) ModelFinder: Fast model selection for accurate phylogenetic estimates. Nature Methods, 14:587–589.) and automatic model selection (Diep Thi Hoang, Olga Chernomor, Arndt von Haeseler, Bui Quang Minh, and Le Sy Vinh (2018) UFBoot2: Improving the ultrafast bootstrap approximation. Mol Biol Evol, 35(2):518-522.)

Comment:

Line 147 - Please use commas in numbers.

Response:

We have changed this in the manuscript accordingly.

Comment:

Lines 159-160 - Did you have contigs containing both MCP (TerL) and rRNA genes?

Response:

The dataset did not have such contigs containing both viral hallmark genes and ribosomal subunit genes. rRNA gene containing contigs was particularly large, e.g., more than 1Mb. These contigs were particularly interesting to us as they were listed as 'high quality' in the CheckV output. However, it was already clear from the CheckV output that they were unlikely to be viruses, as the number of viral genes was generally very low (<5), whereas several hundred host genes were predicted. Nevertheless, we screened these contigs manually to be sure and did not identify any viral hallmark genes. Instead, they were genes in the 'other' category.

Comment:

Lines 201-202 - "whereas 1-SewaA would form one genus together with an uncultured Caudoviricetes phage (GenBAnk Acc. No. LR797474)." - It would be interesting and quick to check this assumption of VICTOR with the help of  VIRIDIC or orthoANI (70% genus threshold)

Response:

Thank you very much for this suggestion. We have compared the two genomes using VIRIDIC. The two genomes had 20.7% similarity. As a result, we changed that part in the results section to: "…whereas 1-SewaA would form one genus together with an uncultured Caudoviricetes phage (GenBAnk Acc. No. LR797474). To test this hypothesis, we calculated the intergenomic similarity of these two phages using VIRIDIC with 70% genus threshold [83]. The two genomes were 20% identical, suggesting that the two phages are more distantly related than suggested by the VICTOR analyses."

Comment:

Line 210 - Caudoviricetes should be written in italics

Response:

We have searched the manuscript and italicized all the entries for Caudoviricetes.

Comment:

Line 215 - Bacillus should be written in italics

Response:

We italicized Bacillus.

Comment:

Line 242 - 81% of the hypothetical proteins are normal for phages, but this is not a small amount.

Have you used HHpred (HHblits) for the annotation procedure?

Response:

Yes. We included HHPred searches in the hypothetical protein analysis. Please see line 102 in the Methods section.

Comment:

Line 250 - Please explain the AMG acronym here when you use it for the first time.

Response:

We have changed this in the manuscript accordingly.

Round 2

Reviewer 2 Report

Comments and Suggestions for Authors

L48. 'Their' rather than 'There'.

L52. ‘their replication is independent of the host’ would be better to be revised as ‘their replication is in part independent of the host’. Viruses are dependent on hosts for replication.

L152. Space after ref 70 needed.

Author Response

Dear Reviewer 2,

The authors thank you for the review as we believe details are important in a publication.

We made all the changes requested by the reviewer and this can be seen in the highlighted version of the updated manuscript (pdf).

Kind regards,

Ulisses Rocha